# 17β-Estradiol (E2) Activates Matrix Mineralization through Genomic/Nongenomic Pathways in MC3T3-E1 Cells

**DOI:** 10.3390/ijms25094727

**Published:** 2024-04-26

**Authors:** Hiraku Suzuki, Yuki Fujiwara, Winda Ariyani, Izuki Amano, Sumiyasu Ishii, Ayane Kate Ninomiya, Seiichi Sato, Akinori Takaoka, Noriyuki Koibuchi

**Affiliations:** 1Department of Integrative Physiology, Gunma University Graduate School of Medicine, 3-39-22 Showa-machi, Maebashi 371-8511, Gunma, Japan; hsuzu0703@igm.hokudai.ac.jp (H.S.); y-fujiwara@gunma-u.ac.jp (Y.F.); winda@gunma-u.ac.jp (W.A.); iamano-lj@umin.ac.jp (I.A.); sishii@gunma-u.ac.jp (S.I.); m1820602@gunma-u.ac.jp (A.K.N.); 2Division of Signaling in Cancer and Immunology, Institute for Genetic Medicine, Hokkaido University, Kita-15, Nishi-7, Kita-ku, Sapporo 060-0815, Hokkaido, Japan; seisato@igm.hokudai.ac.jp (S.S.); takaoka@igm.hokudai.ac.jp (A.T.); 3Molecular Medical Biochemistry Unit, Biological Chemistry and Engineering Course, Graduate School of Chemical Sciences and Engineering, Hokkaido University, Sapporo 060-0815, Hokkaido, Japan

**Keywords:** 17β-estradiol (E2), osteoblast, MC3T3-E1, matrix mineralization, estrogen receptor (ER), osteoporosis

## Abstract

Estrogen plays an important role in osteoporosis prevention. We herein report the possible novel signaling pathway of 17β-estradiol (E2) in the matrix mineralization of MC3T3-E1, an osteoblast-like cell line. In the culture media-containing stripped serum, in which small lipophilic molecules such as steroid hormones including E2 were depleted, matrix mineralization was significantly reduced. However, the E2 treatment induced this. The E2 effects were suppressed by ICI182,780, the estrogen receptor (ER)α, and the ERβ antagonist, as well as their mRNA knockdown, whereas Raloxifene, an inhibitor of estrogen-induced transcription, and G15, a G-protein-coupled estrogen receptor (GPER) 1 inhibitor, had little or no effect. Furthermore, the E2-activated matrix mineralization was disrupted by PMA, a PKC activator, and SB202190, a p38 MAPK inhibitor, but not by wortmannin, a PI3K inhibitor. Matrix mineralization was also induced by the culture media from the E2-stimulated cell culture. This effect was hindered by PMA or heat treatment, but not by SB202190. These results indicate that E2 activates the p38 MAPK pathway via ERs independently from actions in the nucleus. Such activation may cause the secretion of certain signaling molecule(s), which inhibit the PKC pathway. Our study provides a novel pathway of E2 action that could be a therapeutic target to activate matrix mineralization under various diseases, including osteoporosis.

## 1. Introduction

Estrogen plays an important role in bone metabolism. Decreasing levels of estrogen in postmenopausal women increases the risk of osteoporosis [1,2,3]. Bone structure is maintained by via remodeling through bone formation and resorption [4,5]. Estrogen is well known to inhibit bone resorption via osteoclast [1,2,6]. Thus, estrogen therapy (ET) decreases the bone fracture and resorption via this process [1]. Recently, however, the mainstream treatment shifted to the anti-resorptive drugs [1,7,8] because of the increased risk of heart attacks and breast cancer from ET [1,2]. Although certain drugs, such as bisphosphonate, also effectively reduce fracture rates and increase bone density, they also present a risk of serious side effects, including atypical femur fractures [2,7] and osteonecrosis of the jaw [2,8]. For the development of osteoporosis treatments without such adverse effects, it may not be a bad idea to reconsider ET in order to accelerate bone formation.

Estrogen (17β-estradiol, E2) regulates the function not only of osteoclast, but osteoblast bone-forming cells [4,6]. E2 regulates the expression of proteins involved in osteoblast functions to promote the osteoprotective effect, including the inhibition of the receptor activator of nuclear factor-κB ligand (RANKL), which activates osteoclast differentiation and function [9,10], and the stimulation of Sema3a, which promotes osteocytes survival [11]. Furthermore, E2 is reported to induce the cell differentiation of bone-forming cells in vitro, such as osteoblasts and mesenchymal progenitor cells (MPC), through bone morphogenetic proteins (BMPs) or Wnt/β-catenin signaling [12,13,14,15,16]. The activation of the Wnt/β-catenin pathway is involved in the mRNA transcription of alkaline phosphatase (ALP) [13,17], which is essential in the early phases of matrix mineralization and several other signaling proteins, such as the WNT1-inducible signaling pathway protein 2 (WISP2), which is involved in MPC differentiation [18,19,20]. E2 may influence the Wnt/β-catenin system by inhibiting sclerostin, which then inhibits Wnt signaling [20]. Furthermore, E2 triggers PKC and p38 MAPK signaling, which are important for the differentiation of bone-forming cells [21,22,23,24]. Although the mechanisms triggering PKC and p38 MAPK signaling via E2 has not yet been clarified, E2 may activate these systems under non-genomic pathways [25]. Considering such important roles of E2 on bone-forming cells, it may be important to examine the involvement of these systems on the matrix mineralization process in addition to the cell differentiation process.

As stated above, there are so many reports on the action of E2 in bone-forming cells. Nevertheless, the mechanisms of E2 action on the matrix mineralization process has not yet been fully clarified. E2 may act through multiple pathways in a differentiation stage-dependent manner [5,13,14,15,21]. In this study, we focused its effect on the autocrine/paracrine loop of signaling in bone-forming cells [26,27] using cultured MC3T3-E1 osteoblast precursor cells with “stripped” fetal bovine serum (FBS), in which lipophilic materials, such as steroid hormones, including E2, was depleted by charcoal treatment [13]. We also used a phenol red free medium, because phenol red may induce estrogenic activity [28]. In using this phenol red free medium with a stripped serum, we could precisely examine the effect of E2 on the differentiation/matrix mineralization process of MC3T3-E1 cells, and we found possible involvement of the autocrine/paracrine pathways of this process.

## 2. Results

### 2.1. E2 Stimulation Triggers Matrix Mineralization of MC3T3-E1 Cells

To determine the effects of E2 on bone formation, we first investigated whether E2 stimulation influenced the matrix mineralization of MC3T3-E1 cells in the culture medium with a stripped serum (Figure 1). We confirmed that E2 was almost completely excluded in the stripped serum (Appendix A). Under this condition, we cultured MC3T3-E1 cells with E2 from Day 0 to Day 3, with 50 mg/mL L (+)-ascorbic acid (AA) and 10 mM β-glycerophosphate (βGP, Sigma) from Day 0 to Day 12. A significant increase in matrix mineralization was observed with 1–100 nM of E2 stimulation on Day 12 (Figure 2A). Furthermore, E2 stimulation also induced a significant increase in alkaline phosphatase (ALP) activity in the culture media on Day 12 (Figure 2B) and Day 9 (Appendix A). On the other hand, matrix mineralization on Day 12 in the standard culture media (no stripping) was observed, regardless of E2 stimulation (Appendix A).

Next, we examined the change in the mRNA levels of differentiation marker proteins via RT-PCR. We cultured this with or without E2 from Day 0 to Day 3, and then harvested on Day 3, Day 6, and Day 12. Without E2 treatment, no significant changes in mRNA levels were observed in all of the examined genes. On the other hand, E2 treatment induced a significant increase in *Col1a* mRNA on Day 3, *Alp* mRNA on Day 3, 6, and 12, and *Ocn* mRNA on Day 6 and 12 (Figure 3).

### 2.2. E2 Stimulation Induces the Matrix Mineralization of MC3T3-E1 via the ER Pathway Independent from Nuclear Translocation and Transcription Regulation

It is reported that E2 binding to its receptors activates two signaling pathways, the genomic and the non-genomic pathway [29,30,31]. The estrogen receptor (ER) belongs to the nuclear receptor superfamily and is encoded by two genetic loci, which produce ERα and ERβ, which then regulate the transcription of the target gene in a ligand-dependent manner [12]. Estrogen also binds to a G protein-coupled receptor 30 (GPR30), also termed as a G protein-coupled estrogen receptor (GPER1), which regulates cell function through activating intracellular signal transduction pathways [25]. Since a previous study indicates the possible involvement of E2 via GPR30/GPER1 on the differentiation and proliferation of bone-forming cells [32], we examined the involvement of GPR30/GPER-mediated pathways on matrix mineralization in MC3T3-E1, as well as the involvement of ER-mediated pathways using their specific inhibitors [1,2,31]. The treatment of ICI182,780, an inhibitor of ERα/β, from Day 0 to Day 3 suppressed the matrix mineralization induced by 10 nM E2 stimulation (Figure 4A). On the other hand, the inhibition of ERα/β translocation into the nucleus with Raloxifene did not significantly inhibit E2-induced matrix mineralization (Figure 4B). The treatment of G15, an antagonist of GPR30/GPER1, also did not induce a significant change on E2-induced matrix mineralization (Figure 4C). To determine the mechanism causing such differences between ICI182,780 and Raloxifene, we measured the ALP activity in the culture media and mRNA levels of *Ocn* with E2 and the ER inhibitors. Although the E2-induced increase in the ALP activity in the culture media was significantly decreased by ICI 182,780, the ALP activity was still significantly upregulated by E2 (Figure 5A, left). On the other hand, the E2-induced increase in the ALP activity in the culture media was completely suppressed by Raloxifene (Figure 5A, right), although the matrix mineralization was not affected by its treatment. An increase in the *Ocn* mRNA level, caused by E2, was significantly suppressed by ICI182,780 (Figure 5B, left), whereas Raloxifene did not alter the effect of E2 (Figure 5B, right). These results indicate that the nuclear event of ER which was suppressed by Raloxifene may not be mainly involved in matrix mineralization, although it may contribute to the mineralization by regulating ALP secretion into the medium. Since ER also regulates the p38 MAPK pathway [25,33], we examined the phosphorylation of p38 MAPK. The E2 treatment for 1 h on Day 0 showed the nuclear translocation of ERs. We also observed the increased phosphorylation of 38 MAPK, induced by 1 h of E2 stimulation, and it was disrupted by co-incubation with ICI182,780 (Figure 5C).

Moreover, we determined the effects of ERα and ERβ knockdown on matrix mineralization. Significant decreases were observed in ERα and ERβ mRNA levels, induced by each specific siRNA (Figure 6A). The knockdown of either ER suppressed E2-induced matrix mineralization on Day 9 (Figure 6B).

### 2.3. Inhibition of the p38 MAPK Pathway and the Activation of the PKC Pathway, Excluding the PI3K Pathway, Suppressed E2-Stimulated Matrix Mineralization

The differentiation of bone-forming cells including MC3T3-E1 are regulated by the activation and inactivation of many signaling pathways [4,5]. As shown above, E2 may not activate through nuclear signaling, but may activate the p38 MAPK pathway in order to induce matrix mineralization. Furthermore, we have tested the involvement of other pathways that may control matrix mineralization. To examine such possibilities, we used SB202190, an inhibitor of the p38 MAPK pathway, PMA, an activator of PKC, and Wortmannin, a PI3K inhibitor. These pathways have been reported to play important roles in the differentiation of bone-forming cells [34,35,36]. A significant suppression of E2-mediated matrix mineralization was observed to be induced by SB202190 (Figure 7A). A significant suppression was also observed via PKC activation by PMA (Figure 7B). On the other hand, E2 action was not inhibited by PI3K inhibition with Wortmannin (Figure 7C). Together with the results in Figure 5C, the p38 MAPK pathway may play an important role in E2-mediated matrix mineralization. The PKC pathway may be also involved.

### 2.4. E2 Stimulation Induces p38 MAPK Activation, Which May Inhibit PKC Pathway in an E2- Independent Manner

As shown in Figure 3, we observed the changes in mRNA levels mainly on Day 6 and 12, although E2 treatment occurred from Day 0 to Day 3, followed by us changing the media that does not contain E2. This finding led us to speculate that some unidentified factors might be secreted by E2 treatment and modulate matrix mineralization. To clarify this speculation, we exposed 10 nM E2 for 6, 12, or 18 h, followed by the replacement of the medium that did not contain E2 (Figure 8A). Then, these E2-free replaced media from E2 pre-exposed cells were collected and applied into new cultures. The culturing continued for 3 days. Then, the culture media were changed again and incubated for an additional nine days (total twelve days), with the culture media changing every three days. Cultures that were directly exposed to E2 for 6, 12, or 18 h also continued for 12 days. As shown in Figure 8B, a significant matrix mineralization was observed in the cells incubated with the replaced E2-free media after E2 stimulation for 18 h, as seen in the cells that were directly incubated with E2 for 18 h. The replaced E2-free media after 6 or 12 h of E2 stimulation also showed the same tendency. However, the degree of matrix mineralization varied greatly among cultures, and thus a clear effect of the replaced media was not determined. However, these results indicate that it may not be E2 itself, but something else that was secreted upon the stimulation of E2 and that activated matrix mineralization.

Furthermore, we investigated which signal transduction pathway is activated by the E2-stimulated factor to activate matrix mineralization. To test this, as shown in Figure 9A, we stimulated Day 0 MC3T3-E1 with E2-free media from the E2 pre-exposed cell. First, we stimulated MC3T3-E1 with 10 nM E2 and changed the culture medium after 18 h of stimulation. After 6 h of incubation, we collected the media (Figure 9A, upper) and applied it to the culture from Day 0 to Day 3 with PMA or SB202190. We also treated the culture media with heat (95 °C, 3 min, followed by centrifuge at 14,000× *g*, for 3 min) and applied it in the culture (Figure 9A, lower). While PMA significantly suppressed matrix mineralization on Day 12, as seen in directly E2 stimulated cells (Figure 9B), SB202190 did not influence culture media-induced matrix mineralization (Figure 9C), indicating that the PKC pathway was affected by the E2-stimulated factor, but the MAPK pathway was not. Finally, we found that the heat treatment significantly extinguished the effects of the culture media (Figure 9D). A significant increase in the *Ocn* mRNA level, induced by the culture media from E2 pre-exposed cells on Day 12, was also suppressed by heat treatment (Figure 9E).

## 3. Discussion

In this study, we showed a novel signaling pathway of E2 on the matrix mineralization of bone-forming cells using the MC3T3-E1 cell, a mouse-derived pre-osteoblast clonal cell. Our study demonstrated that matrix mineralization may not be induced by E2 itself, but by the paracrine signaling pathway activated by E2. As shown in the graphical abstract, E2 may first activate the p38 MAPK pathway, leading to the secretion of a yet unidentified factor, which then inhibits the PKC pathway, causing matrix mineralization. Although such factors have not yet been identified, out study may contribute greatly to further understanding the role of E2 in terms of matrix mineralization, and to providing novel therapeutic targets in order to treat diseases such as post-menopausal osteoporosis.

Bone formation is regulated by the activation/inhibition of various signaling pathways and signaling molecules [5]. There are many reports on estrogen action in bone-forming cells [10,11,12,13,14]. However, according to the previous reports, the culture medium used in vitro assays in some studies were constructed with αMEM and FBS, which contain a detectable amount of E2 and other estrogens. By using FBS, the exact effects of estrogen may not be able to be examined in detail. The “stripping” of serum is a common technique in the research area of endocrinology. Using this method, estrogens and other lipophilic hormones, which may also synergistically affect estrogen-mediated pathways, such as progesterone, can be eliminated (Appendix A), thus enabling us to examine the true estrogen action. By using stripping serum, we were able to detect a novel E2-mediated signaling pathway in the matrix mineralization of bone-forming cells.

We found that matrix mineralization rarely occurred in MC3T3-E1 for 12 days after the stimulation of differentiation in the culture medium containing stripped FBS. However, only 3 days of stimulation using 1–100 nM of E2 induced matrix mineralization (Figure 2A), accompanied by an increase in ALP activity in the culture medium (Figure 2B). Matrix mineralization induced by E2 was suppressed by ICI182,780, an ER antagonist (Figure 4A), and the knockdown of either ERα or ERβ (Figure 6B), whereas G15, a selective antagonist of GPR30/GPER1, did not influence the E2-induced effect (Figure 4B), indicating the involvement of ERs. On the other hand, Raloxifene, one of the selective estrogen receptor modulators (SERM), did not suppress E2-induced matrix mineralization (Figure 4C), although it completely disrupted the increase in ALP activity in the culture medium (Figure 5A, right). The E2-induced expression of *Ocn* mRNA was significantly inhibited by ICI182,780, whereas Raloxifene did not induce any changes (Figure 5B). Although both ICI182,780 and Raloxifene antagonize ER action, there is a difference in the mechanism of action. ICI182,780 inhibits both genomic and non-genomic ER-mediated action [37]. On the other hand, SERM (including Raloxifene)-bound ERs translocate into the nucleus, but did not bind to nuclear receptor coactivators, thus inhibiting the transcription of target genes [38]. Thus, the results of the present study lead us to speculate that the non-genomic pathway of estrogen signaling may be mainly involved in E2-mediated matrix mineralization.

Based on this hypothesis, we examined the phosphorylation of p38 MAPK via E2 exposure for 1 h. E2 treatment significantly increased the phosphorylation of p38 MAPK, whereas ICI182,780 inhibited it. Based on this result, we considered that this phosphorylation was induced by non-genomic E2 action through extranuclear ER, because Raloxifene only weakly affected the matrix mineralization induced by E2.

Furthermore, we treated cells with E2 for 6, 12, or 18 h, followed by a medium change. We have collected such replaced media that do not contain E2 and applied it to other cultures for 3 days. We found a significant increase in matrix mineralization with the replaced media collected from 18 h exposed cultures. Treatment with ICI182,780 did not affect the matrix mineralization in this culture. These results indicate that an unidentified factor, which was induced by E2, was secreted into the medium and activated matrix mineralization. The effect of the culture media was extinguished by 95 °C heat treatment, indicating further that a higher structure of this factor (could be protein or peptide) was destroyed, and bioactivity was lost via heat treatment. On the other hand, the effect of the replaced culture media from E2-treated cells were inhibited by PMA, indicating that the PKC pathway is located in the downstream of the E2-induced secretory factor signal transduction pathway. In addition, we observed that SB202190 did not affect matrix mineralization induced by the replaced medium, indicating that the p38 MAPK pathway is located upstream of the E2-meidiated secretory factor. Taken together, our results indicate that E2 activates the p38 MAPK pathway through the non-genomic ER pathway, causing the secretion of unknown signaling molecules into the culture medium. This factor may play a major role in causing matrix mineralization.

Although the substances which play a major role in matrix mineralization in our preparation were not fully identified, Ocn and alkaline phosphatase may be involved in this process. While the E2-induced significant increase in *Ocn* mRNA levels was suppressed by ICI182,780, Raloxifene did not alter the effect of E2 (Figure 5B), indicating that the genomic pathway may not be involved. Since Ocn plays a major role in the matrix mineralization process, it is reasonable to speculate that Ocn expression is activated though the p38 MAPK-mediated pathway, most likely in the downstream of the secreted protein-mediated pathway. Regarding ALP, on the other hand, although ICI182,780 partially suppressed E2-activated ALP activity in the culture medium, Raloxifene completely blocked it, indicating that ALP activity in culture media is mainly regulated by nuclear ER action. However, the regulation of ALP levels may not play a major role in E2-mediated matrix mineralization, as discussed above. The results of the present study indicate that it could be Ocn, not ALP, which plays a major role in regulating E2-activiated matrix mineralization. However, possibilities of the involvement of additional factors cannot be excluded. Further study may be required in this direction.

Although we have clarified the involvement of the ER-mediated non-genomic pathway in our study, there are additional questions to be addressed. Particularly, we have found that the knockdown of either ERα or β is sufficient to inhibit E2 action. It is quite an unexpected finding, and, at present, we do not have a clear answer for this finding. The mechanism of ER action in the plasma membrane and cytosol are not fully clarified. Under such a condition, we do not want to make a speculative comment regarding our findings. However, based on our study, the interaction between ERα and β may be essential in regulating E2-mediated matrix mineralization.

In conclusion, we showed that E2 induced matrix mineralization by interacting with ERs in MC3T3-E1 cells, most likely through the p38 MAPK-mediated non-genomic pathway. Furthermore, we have shown the possibility that this E2-madiated matrix mineralization could be mainly induced by an unidentified factor which is secreted into the culture media via the activation of E2 signaling. This secretory factor inhibited the PKC pathway, causing matrix mineralization. Although factors that directly contribute to the E2-activated mineralization process were not fully identified, Ocn can be a potent factor that is expressed by the stimulation of the secretory factor. Our study has provided a novel pathway of the estrogen-mediated ossification process. Such findings may contain clinical relevance to identifying novel therapeutic targets for bone diseases induced by estrogen deficiency, such as osteoporosis.

## 4. Materials and Methods

### 4.1. Reagents

E2 (17β-estradiol) and G15 ((3aS,4R,9bR)-4-(6-bromo-1,3-benzodioxol-5-yl)-3a,4,5,9b-tetrahydro-3H-cyclopenta[c]quinoline) were purchased from CAY. ICI 182,780 (7α,17β-[9-[(4,4,5,5,5-Pentafluoropentyl)sulfinyl] nonyl] estra-1,3,5(10)-triene-3,17-diol) was from RSD. Wortmannin, PMA, and SB202190 were from Sigma (St. Louis, MO, USA).

### 4.2. Generation of the Stripped Serum

FBS (biowest) was heat inactivated for 30 min at 56 °C. It was stirred overnight with resin (Bio-Rad, Hercules, CA, USA, AG1x8 Resin 200–400 mesh, chloride form), followed by stirring with the resin and charcoal (Mallinckrodt, activated, Hazelwood, MO, USA) for 4 h. After centrifuging at 2000 rpm for 10 min at 4 °C and 10,000 rpm for 20–30 min at 4 °C, the supernatant was sterilized via filtration. It is well known that such treatment can deplete lipophilic materials such as steroid and thyroid hormones from the serum (Appendix A).

### 4.3. Cell Line and Culture

The MC3T3-E1 osteoblast precursor cell line was obtained from the RIKEN Cell Bank (Tsukuba, Japan), and was maintained in Minimum Essential Medium Eagle (MEMα with L-Glutamine, Sodium Pyruvate and Nucleosides, Wako, Osaka, Japan) containing 10% FBS, 50 U/mL penicillin, and 50 mg/mL streptomycin (Wako). Three days before the onset of the cellular differentiation stimulation, the culture media was replaced with the culture media containing 10% stripped FBS, 50 U/mL penicillin, and 50 mg/mL streptomycin. To induce differentiation, we added 50 mg/mL L (+)-ascorbic acid (AA, Wako) and 10 mM β-glycerophosphate (β-GP, Sigma). The culture media was changed every three days. The day when differentiation was induced was designated as Day 0 in the experiment.

To examine the effect of estrogen on the matrix mineralization process, an indicated amount of E2 (1, 10, or 100 nM) was added into the culture medium on Day 0. E2 exposure continued until Day 3, then the culture medium was replaced with an E2-free medium. The medium was changed every 3 days. Then, the culture media were collected for the ALP assay, and the cells were fixed by 10% formaldehyde for Alizarin Red staining. Some cells were harvested on day 0, 3, 6, or 12 to measure the mRNA or protein levels.

Then, to examine the involvement of various receptors, such as ICI 182,780, an ERα, and an β antagonist (for both nuclear and non-nuclear actions) [33], G15, a G-protein-coupled estrogen receptor (GPER)1 antagonist, or Raloxifene, an ER α- and β-mediated transcription inhibitor [31,32], were added at a concentration of 100 nM, respectively, on Day 0 until Day 3. Then, the culture media was collected for the ALP assay. Some cells were harvested for the osteocalcin (Ocn) mRNA measurement using RT-PCR, whereas other cells were fixed by formaldehyde for Alizarin Red S staining.

To examine the phosphorylation of the p38 mitogen-activated protein kinase (MAPK), cells were harvested after 1 h of E2 treatment on Day 0. In other cultures, to examine the involvement of various signal transduction pathways, Wortmannin, a PI3 kinase (K) inhibitor, phorbol myristate acetate (PMA), a protein kinase C activator, or SB202190, a MAP kinase inhibitor, was added at a concentration of 500 nM, 150 nM, or 10 µM, respectively, on Day 0 until Day 3. These cultures were fixed with 10% formaldehyde, followed by Alizarin Red S staining (see below).

To examine the effect of the culture media obtained from E2 pre-exposed cells, cells were cultured for 6, 12, or 18 h with 10 nM E2, from Day 0, followed by the replacement of the E2-free medium, and then additionally cultured for 18, 12, or 6 h, respectively. Then, the medium was collected. These media were then added to other cultures for 3 days from Day 0, followed by additional 8 days of culturing. Some media obtained from E2 18 h exposed cells were added with PMA or SB202190 for 3 days, followed by an additional 8 days of culturing. Another medium obtained from E2 18 h exposed cells were treated with heat (95 °C, 3 min) and centrifuged (4 °C, 14,500 rpm, 5 min) before being added to the culture.

### 4.4. Alizarin Red S Staining

Matrix mineralization was measured using Alizarin Red S (Wako) staining, according to the previous study [26]. The mineralization area of the stained cultures were photographed and measured using Image J software version 1.53 (RRID: SCR_003070). The differences between the control (no estrogen) and experimental groups were shown in relative values. The average value of the control was defined as 1.

### 4.5. Quantitative RT-PCR (qRT-PCR)

Total RNAs were isolated from cultured cells harvested on Day 0, 3, 6, or 12, using ISOGENII (Nippon Gene, Tokyo, Japan). cDNA preparation was performed from total RNAs using ReverTra Ace (TOYOBO, Osaka, Japan). qRT-PCR for mRNAs for ossification-related genes, such as *Alp* (*Alpl*), runt-related transcription factor 2 (*Runx2*), type 1 collagen (*Col1a*), *Ocn*, and reference gene *Rpl13a*, was performed using iTaq Universal SYBR Green Supermix (Bio-Rad and analyzed the StepOnePlus real-time PCR system (Applied Biosystems, Foster City, CA, USA). The data were normalized to the expression of *Rpl13a* for each sample. The value was divided by the average value of the Day 0 samples and the Day 0 value was set to 1. The primer pairs used for PCR are shown in Table 1 [26,27,39].

### 4.6. Alkaline Phosphatase (ALP) Assay

ALP activity in the culture medium was measured using the p-nitrophenol phosphate method with LabAssay^TM^ ALP (Wako). After 15 min of incubation at 37 °C, the reaction was terminated by the addition of a sodium hydroxide solution. The absorbance of p-nitrophenol liberated in the reactive solution was read at 405 nm.

### 4.7. Knockdown of ERs by RNA Interference

Chemically synthesized small interfering (si)RNA for the estrogen receptor (ER)α gene (*ESR1*) (SASI_Mm01_00108712), ERβ gene (*ESR2*) (SASI_Mm01_00185612), and control siRNA (MISSION^®^ siRNA Universal Negative Control #1) were purchased from Sigma. Cells were transfected with 50 nM siRNA in Lipofectamine RNAiMAX (Invitrogen, Carlsbad, CA, USA).

Before the initiation of differentiation, cells were treated with the control or with siRNA for *ESR1* or *ESR2* at concentrations of 10 n M for 48 h. Then, some cells were harvested for the RT-PCR of *ESR1* or *ESR2*, whereas other cells were cultured for 9 days with 10 nM E2, followed by a formaldehyde treatment for Alizarin Red S staining.

### 4.8. Western Blot Analysis

The total proteins from the MC3T3-E1 cells were extracted using a lysis buffer, and the absorbance was measured with Bio-Rad Protein Assay Dye Reagent Concentrate (Bio-Rad). After the addition of the 2xSDS buffer (125 mM Tris-HCl, pH 6.8, 4% SDS, 20% glycerol, 0.04% bromophenol blue, 10% 2-mercaptoethanol) to 1 mg proteins, the samples were heated for 3 min at 100 °C. After 10% SDS-polyacrylamide gel electrophoresis (SDS-PAGE), the samples were developed and transferred to a polyvinylidene difluoride (PVDF) membrane Immobilon-P (Pore size 0.45 mm: GE Healthcare Life Science, Chicago, IL, USA). The transferred PVDF membrane was immersed in 1% BSA TBS-T, containing a 0.6% TBS solution for 1 h at room temperature (RT). After testing the primary antibody (Phospho-p38 MAPK <Thr180/Tyr182> Antibody #9211 and p38 MAPK Antibody #9212, Cell Signaling Technology, Danvers, MA, USA) at 4 °C for overnight, the antibody was washed with TBS-T for 30 min, followed by testing the secondary antibody (Anti-rabbit IgG, HRP-linked Antibody #7074, Cell Signaling Technology) at 4 °C for 1 h, then being washed with TBS-T for 30 min. Chemiluminescence emission was detected using ImmunoStar Zeta (Wako), and the protein band intensities were assessed using LAS 4010 mini (GE healthcare). The values of the band were divided by the average values of the control band set to 1. The ratio of band intensity was measured with ImageJ software version 1.53 (RRID: SCR_003070).

### 4.9. Statistical Analysis

Values were expressed as means ± SD of triplicate independent experiments. Statistical analyses were performed using one-way ANOVA, followed by Bonferroni’s test as a post hoc test with Prism 10. A *p*-value of <0.05 is considered to be significant.

## Figures and Tables

**Figure 1 ijms-25-04727-f001:**
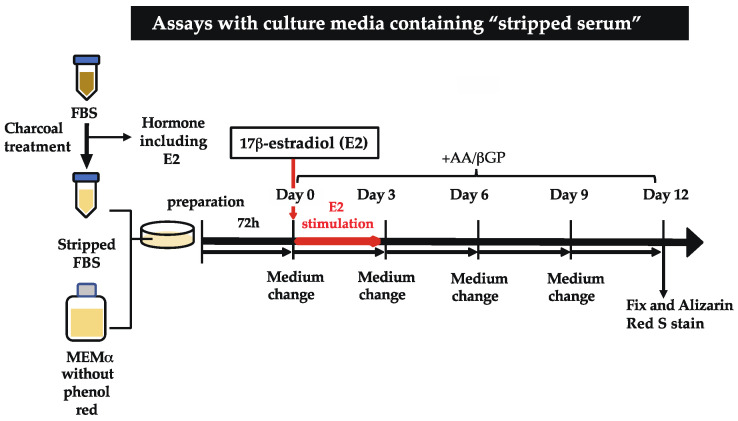
The scheme of matrix mineralization assays used in the present study.

**Figure 2 ijms-25-04727-f002:**
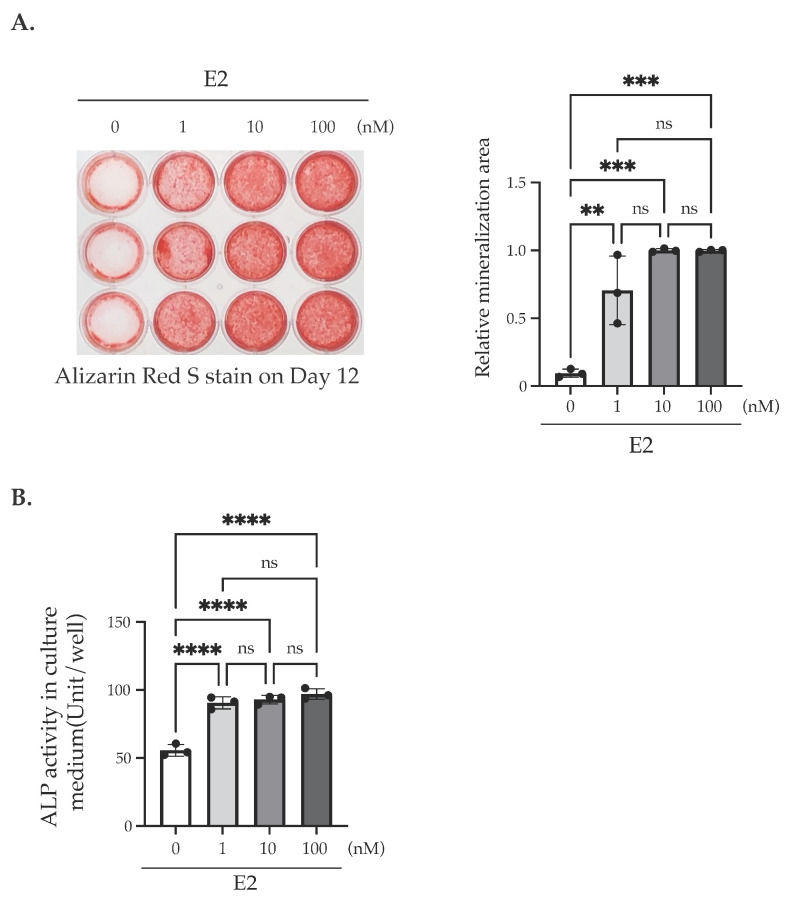
The effect of E2 treatment on the matrix mineralization of MC3T3-E1 cells. (**A**) A representative showing the matrix mineralization assay. Cells were cultured in the media with AA/βGP and cultured for 12 days. The indicated amount of E2 (1–100 nM) was added at Day 0 until Day 3. The cells were fixed and stained using Alizarin red S to determine the matrix mineralization level. The quantification results are shown in the right panel. (**B**) ALP assay. ALP activities in the culture medium on Day 12 of the cells being stimulated with/without E2 (1–100 nM) for 3 days (Day 0–Day 3). Data presented as mean ± SD (*n* = 3) are representative of at least three independent experiments. ** *p* < 0.01, *** *p* < 0.001, and **** *p* < 0.0001 versus control (no E2) using Bonferroni’s test. ns: not significant.

**Figure 3 ijms-25-04727-f003:**
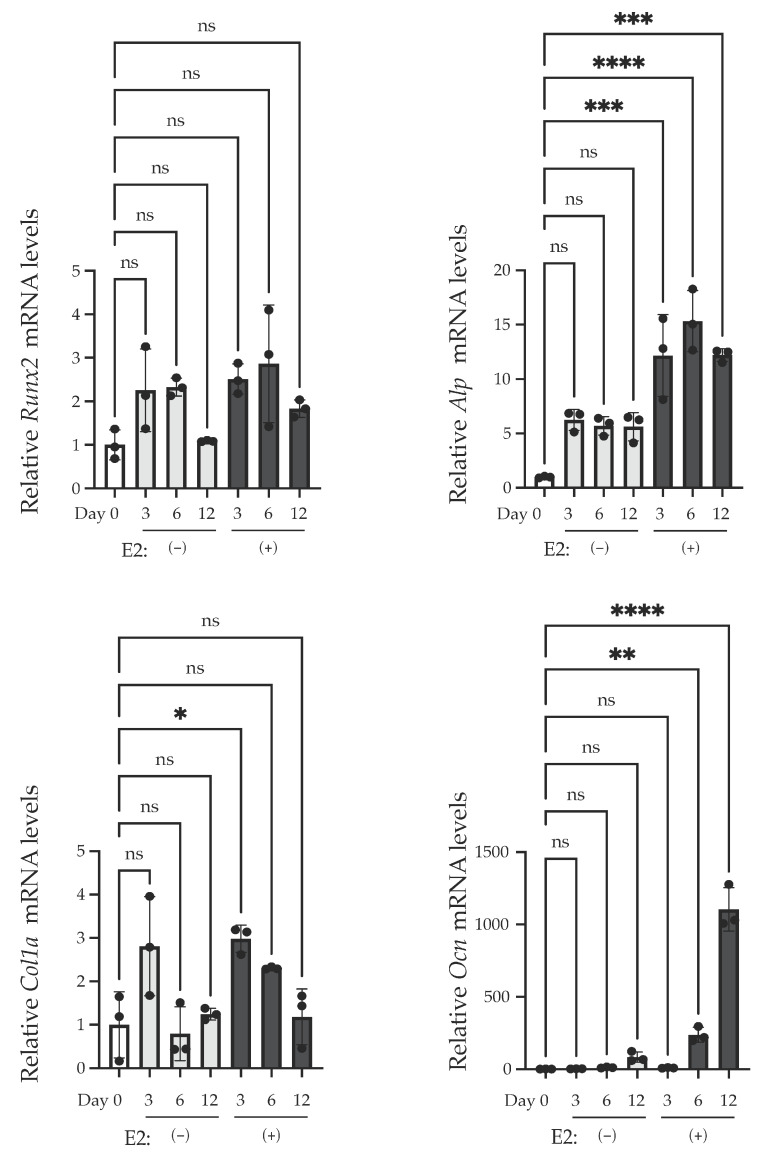
Changes in the mRNA levels of the substance responsible for ossification via E2 treatment. Quantitative RT-PCR analysis for *Runx2*, *Alp* (*Alpl*), *Col1a,* and *Ocn* (*Bglap*) mRNA. mRNA was isolated from MC3T3-E1 cells on the indicated day. E2 (10 nM) was treated for 3 days (Day 0–Day 3). The relative mRNA levels of *Runx2*, *Alp* (*Alpl*), *Col1a*, and *Ocn* (*Bglap*) were normalized to the expression of *Rpl13a*. Data presented as mean ± SD (*n* = 3) are representative of at least three independent experiments. * *p* < 0.05, ** *p* < 0.01, *** *p* < 0.001, and **** *p* < 0.0001 using Bonferroni’s test. ns: not significant.

**Figure 4 ijms-25-04727-f004:**
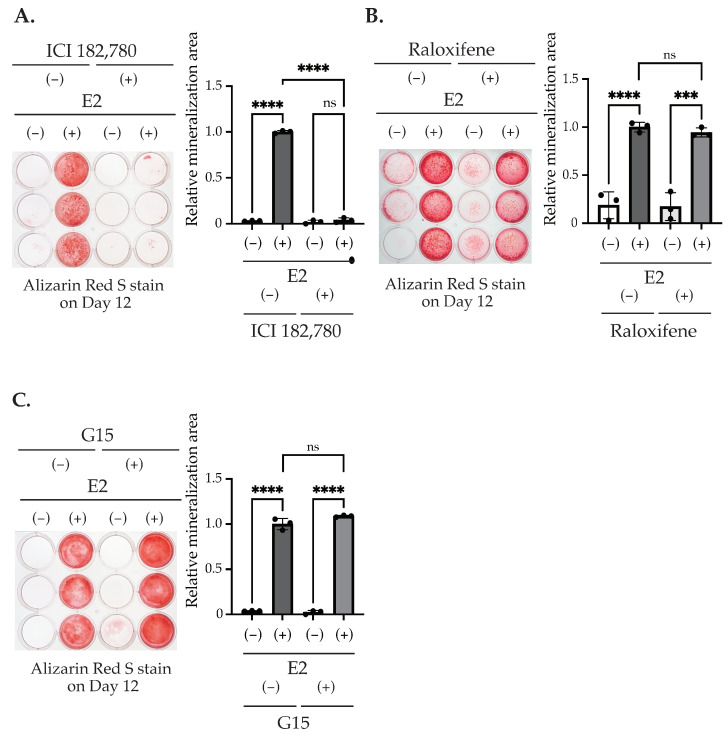
The effect of ER or GPER antagonists on matrix mineralization. (**A**–**C**) MC3T3-E1 cells were treated with/without 10 nM E2 and 100 nM ICI182,780, 100 nM G15, or 100 nM Raloxifene for 3 days (Day 0–Day 3). The cells were fixed and stained with Alizarin red S on Day 12 to determine the matrix mineralization level. The quantification results are shown in the right panel. Data presented as mean ± SD (*n* = 3) are representative of at least three independent experiments. *** *p* < 0.001, **** *p* < 0.0001 using Bonferroni’s test. ns: not significant.

**Figure 5 ijms-25-04727-f005:**
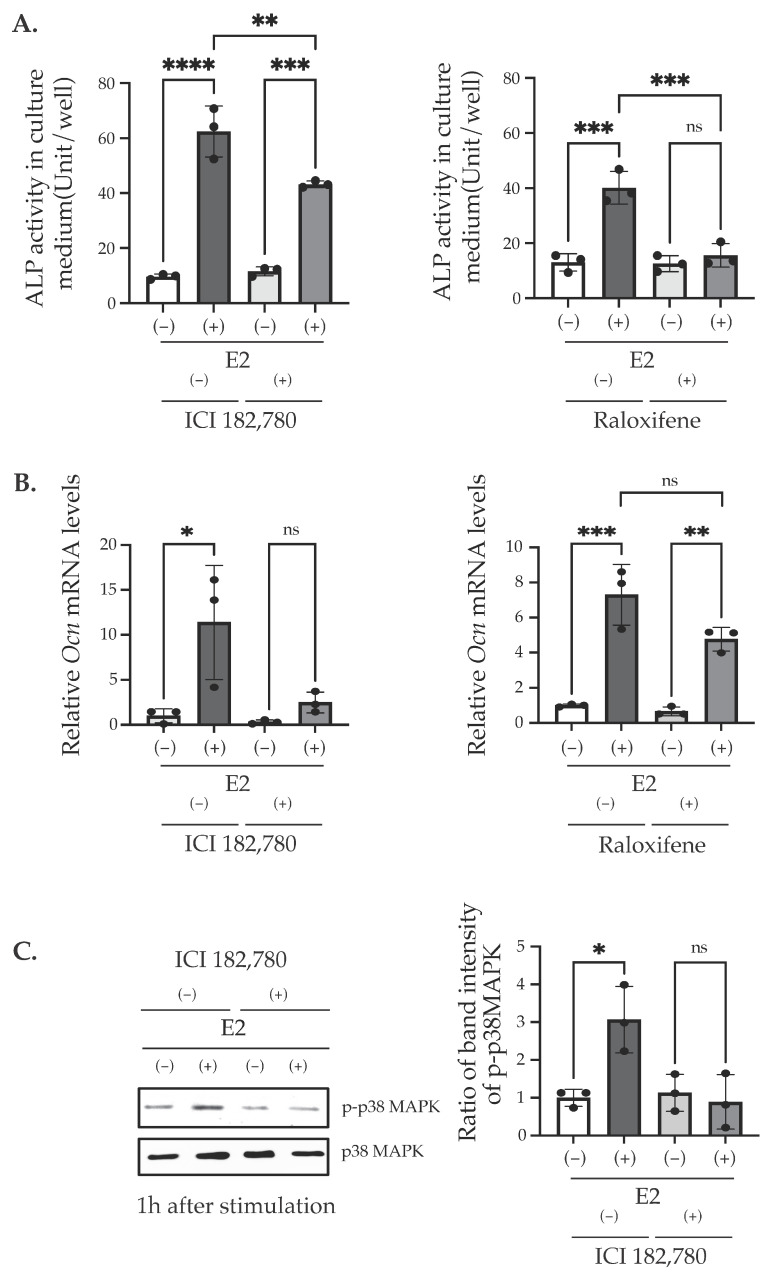
The effects of ER antagonists on ALP activity in the culture media, *Ocn* mRNA, and p38 MAPK phosphorylation. (**A**) ALP assay. ALP activities in the culture medium on Day 12 of the cells treated with/without E2 (10 nM) for 3 days (Day 0–Day 3) in the presence of 100 nM ICI182,780 or 100 nM Raloxifene for 3 days (Day 0–Day 3). (**B**) The quantitative RT-PCR analysis of *Ocn* (*Bglap*). Cells were stimulated with/without E2 (10 nM) in the presence of 100 nM ICI182,780, 100 nM Raloxifene, or not on Day 0. mRNA was isolated from MC3T3-E1 cells on Day 12. Relative mRNA expression was normalized with the expression of *Rpl13a*. (**C**) The Western blot analysis of whole-cell lysates with antibodies for phosphor-p38 MAPK (p-p38 MAPK) or total p38 MAPK. MC3T3-E1 were stimulated with/without 10 nM E2 and 100 nM ICI182,780 on Day 0. Cells were collected 1 h after stimulation. The quantification results are shown in the right panel. Data presented as mean ± SD (*n* = 3) are representative of at least three independent experiments. * *p* < 0.05, ** *p* < 0.01, *** *p* < 0.001, and **** *p* < 0.0001 using Bonferroni’s test. ns: not significant.

**Figure 6 ijms-25-04727-f006:**
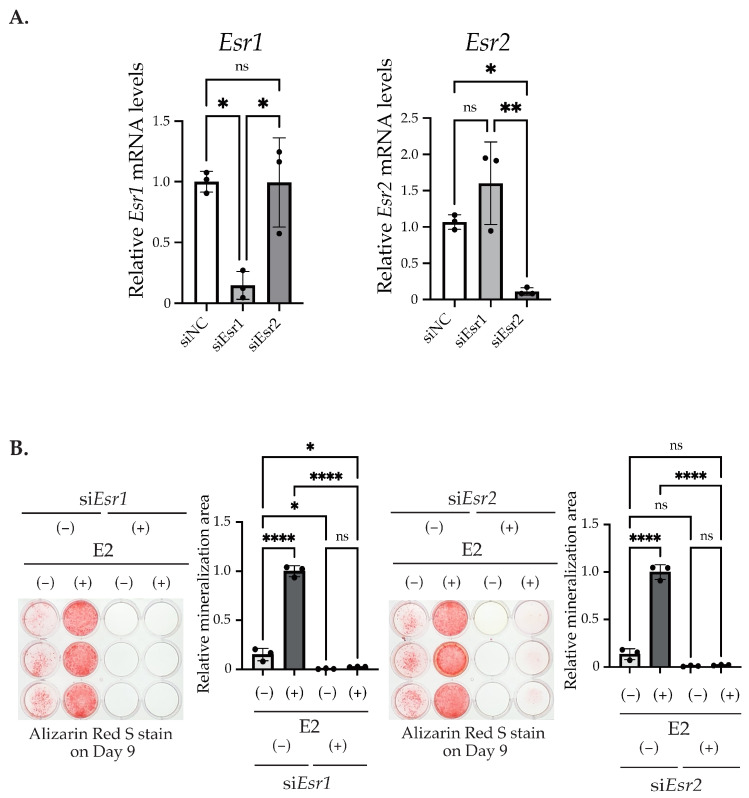
The knockdown of ERα or ERβ during matrix mineralization. (**A**) The quantitative RT-PCR analysis of *Esr1* (ERα) and *Esr2* (ERβ). MC3T3-E1 was treated with control siRNA (Control), si*Esr*1, or si*Esr*2. mRNA was isolated from the cells after 48 h of transfection. Relative mRNA expression was normalized to the expression of *Rpl13a*. (**B**) Matrix mineralization assays. The cells after 48 h of siRNA transfection were stimulated with/without 10 nM E2 for 3 days (Day 0–Day 3). The cells were fixed and stained with Alizarin red S on Day 12 to determine the matrix mineralization level. The quantification results are shown in the right panel. Data presented as mean ± SD (*n* = 3) are representative of at least three independent experiments. * *p* < 0.05, ** *p* < 0.01, and **** *p* < 0.0001 using Bonferroni’s test. ns: not significant.

**Figure 7 ijms-25-04727-f007:**
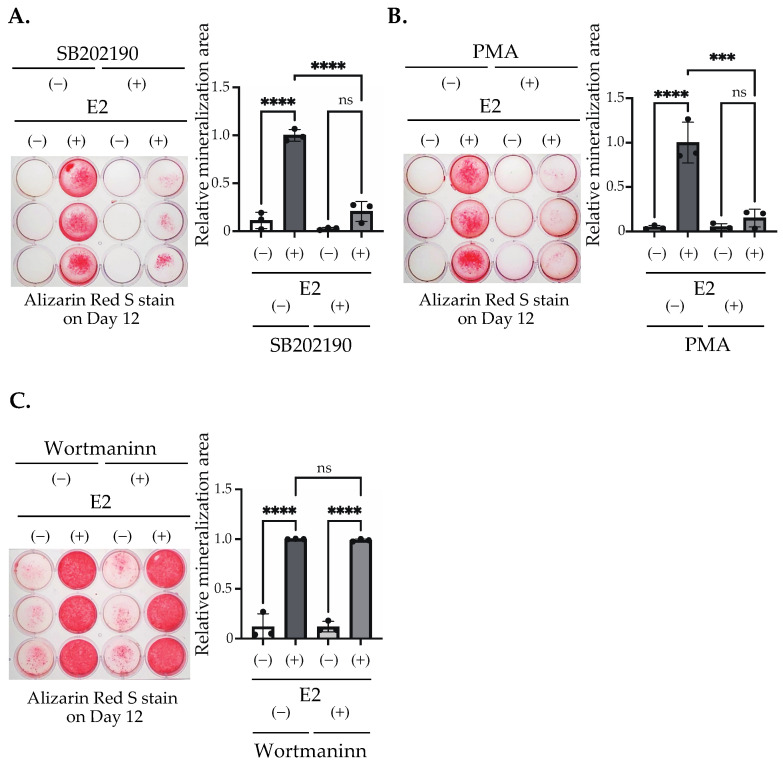
E2 induced matrix mineralization through the inhibition of the PKC pathway and the activation of p38 MAPK pathway. (**A**) Cells were stimulated with/without 10 nM E2 for 3 days (Day 0–Day 3) and cultured with/without (**A**) 500 nM Wortmannin, (**B**) 150 nM PMA, and (**C**) 10 µM SB202190 for 3 days (Day 0–Day 3). The cells were fixed and stained with Alizarin red S on Day 12 to determine the matrix mineralization level. The quantification results are shown in the right panel. Data presented as mean ± SD (*n* = 3) are representative of at least three independent experiments. *** *p* < 0.001, and **** *p* < 0.0001 using Bonferroni’s test. ns: not significant.

**Figure 8 ijms-25-04727-f008:**
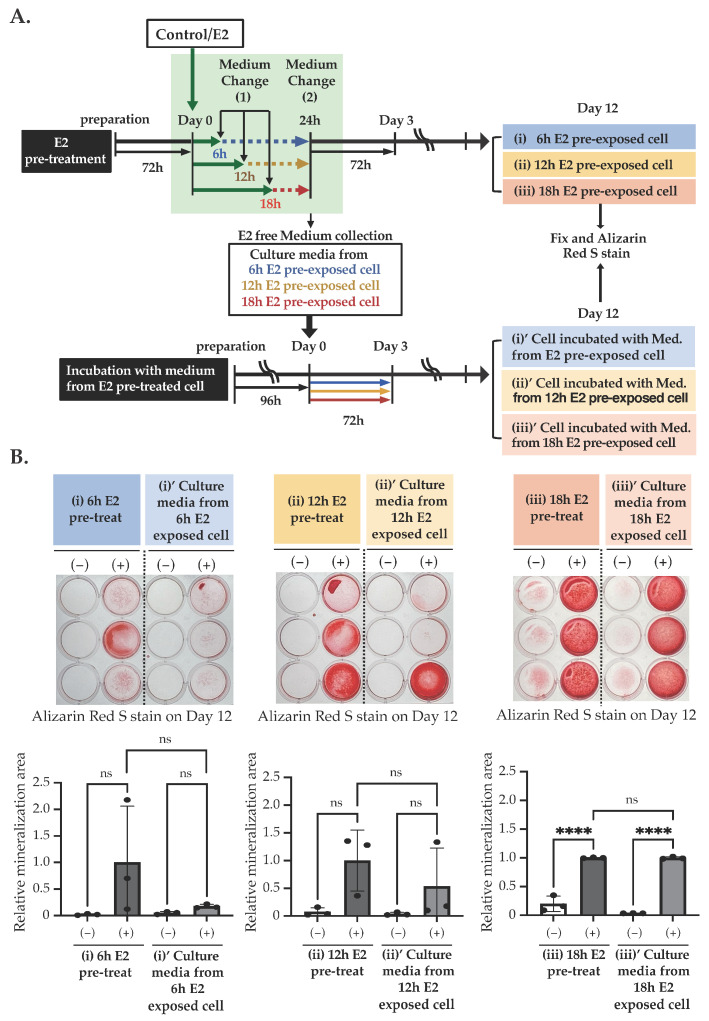
The effect of the culture media obtained from E2 pre-exposed cells on matrix mineralization. (**A**) The scheme of matrix mineralization assays used in the experiments are shown in this figure. (**B**) Matrix mineralization assay. MC3T3-E1 cells were treated with/without 10 nM E2 at Day 0 for 6 h, 12 h, or 18 h, followed by the E2-free medium replacement and incubated for 18 h, 12 h, or 6 h, respectively. Then, cells directly exposed to E2 for 6 h, 12 h, or 18 h (i, ii or iii, respectively) were fixed on Day 12. Cells treated with the culture medium obtained from the 6 h, 12 h, or 18 h pre-exposed cell (i’, ii’, iii’) were also fixed on Day 12. Then, cells were stained with Alizarin red S to determine the matrix mineralization level. Data presented as mean ± SD (*n* = 3) are representative of at least three independent experiments. **** *p* < 0.0001 using Bonferroni’s test. ns: not significant.

**Figure 9 ijms-25-04727-f009:**
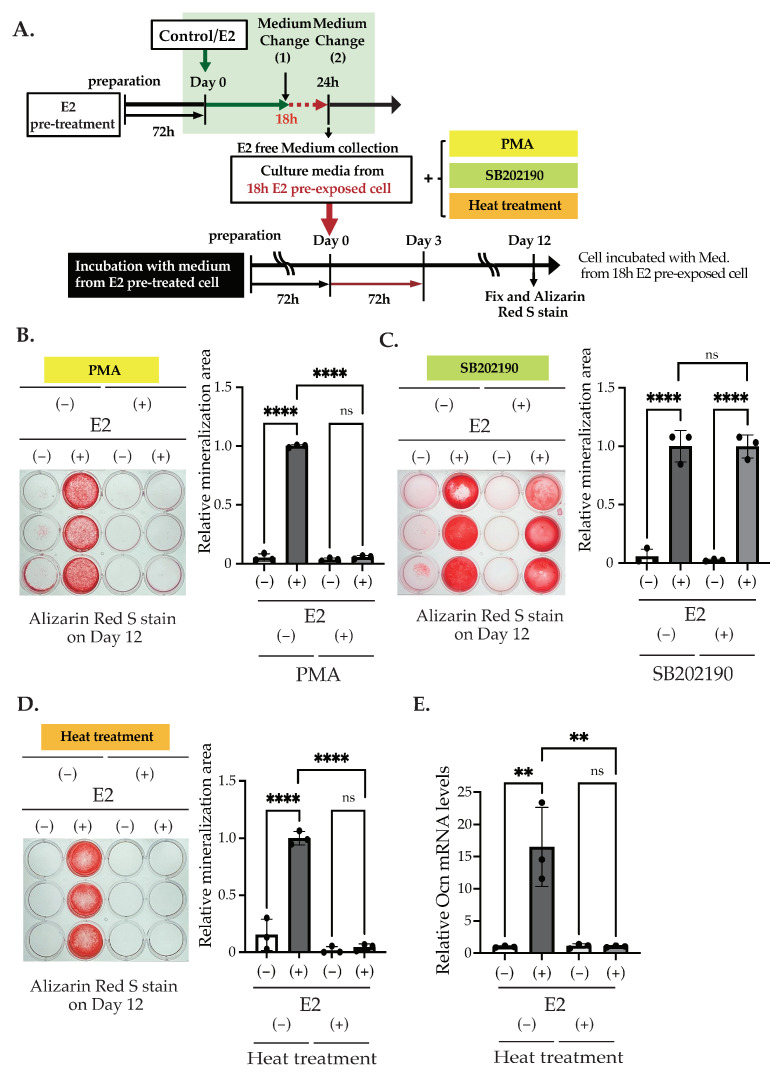
The effect of PKC activation, p38 MAPK inhibition, and heat treatment on matrix mineralization. (**A**) The scheme of matrix mineralization assays used in the experiments is shown in this figure. (**B**,**C**) Matrix mineralization assay. The culture media were collected from the cells pre-exposed with/without 10 nM E2 on Day 0 for 18 h of stimulation. MC3T3-E1 was incubated with the media from the 18 h E2 pre-exposed culture for 72 h in the presence of PMA or SB202190. On Day 12, the cells were fixed and stained with Alizarin red S. The quantification of their results is shown in the right graph. (**D**) Matrix mineralization assay. The collected media were treated with/without 95 °C for 3 min and centrifuged with 14,000× *g*, for 3 min at 4 °C. MC3T3-E1 was incubated with this collected media or the heat-treated media from the 18 h E2 pre-exposed culture for 72 h. On Day 12, the cells were fixed and stained with Alizarin red S. The quantification results are shown in the right panel. The (**E**) quantitative RT-PCR analysis of *Ocn* (*Bglap*). Cells were treated in the same way as in (**D**). mRNA was isolated from MC3T3-E1 cells on Day 12. Relative mRNA expression was normalized to the expression of *Rpl13a*. Data presented as mean ± SD (*n* = 3) is representative of at least three independent experiments. ** *p* < 0.01 and **** *p* < 0.0001 using Bonferroni’s test. ns: not significant.

**Table 1 ijms-25-04727-t001:** Primer sequences.

Gene Symbol (Alias)	Forward Primer	Reverse Primer
*Alp*	ACCTTCTCTCCTCCATCCCT	GTGTGTGTGTGTGTCCTGTC
*Runx2*	GCCCAGGCGTATTTCAGATG	GGTAAAGGTGGCTGGGTAGT
*Col1a*	TGGGCGCGGCTGGTATGAGTTC	ACCCTGCTACGACAACGTGCC
*Ocn*	GCAGAACAGACAAGTCCCAC	ACCTTATTGC CCTCCTGCTT
*Rpl13a*	AGCTTACCTGGGGCGTCTG	ACATTCTTTTCTGCCTGTTTCC

## Data Availability

The original contributions presented in the study are included in the article/Appendix A, further inquiries can be directed to the corresponding authors.

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
