# Peer review of "17β-Estradiol (E2) Activates Matrix Mineralization through Genomic/Nongenomic Pathways in MC3T3-E1 Cells"

_ijms, 2024, doi:10.3390/ijms25094727_

Round 1
Reviewer 1 Report
Comments and Suggestions for Authors
The study uncovers a novel signaling pathway of 17β-estradiol (E2) in matrix mineralization of MC3T3-E1 cells, providing insights into potential therapeutic targets for bone-related diseases. Understanding how E2 activates matrix mineralization through genomic and nongenomic pathways involving estrogen receptors and p38 MAPK activation can contribute to advancements in osteoporosis prevention strategies. The research sheds light on the impact of E2 treatment on differentiation marker proteins like Col1a, Alp, and Ocn mRNA levels, highlighting the significance of estrogen signaling in bone formation.
However, the limitations of this paper are as follows.
1. The study focuses on the specific cell line MC3T3-E1, which may not fully represent the complexity of in vivo bone mineralization processes.
2. The mechanisms of estrogen receptor action in the plasma membrane and cytosol are not entirely clarified, leading to potential uncertainties in the interpretation of results.
3. The use of stripped serum to eliminate lipophilic hormones for studying estrogen action may oversimplify the in vivo hormonal environment, potentially affecting the relevance of the findings.
Comments on the Quality of English LanguageThere is a slight grammatical error.
Author Response
Thank you very much for taking the time to review this manuscript. Please find the detailed responses below.
Point-by-point response to Comments and Suggestions for Authors:
The study uncovers a novel signaling pathway of 17β-estradiol (E2) in matrix mineralization of MC3T3-E1 cells, providing insights into potential therapeutic targets for bone-related diseases. Understanding how E2 activates matrix mineralization through genomic and nongenomic pathways involving estrogen receptors and p38 MAPK activation can contribute to advancements in osteoporosis prevention strategies. The research sheds light on the impact of E2 treatment on differentiation marker proteins like Col1a, Alp, and Ocn mRNA levels, highlighting the significance of estrogen signaling in bone formation.
However, the limitations of this paper are as follows.
Comments 1: The study focuses on the specific cell line MC3T3-E1, which may not fully represent the complexity of in vivo bone mineralization processes.
Response 1: Thank you for pointing this out. We agree with this comment. However, in this study, we needed to construct simplified experimental model as a first step to find new estrogen function in bone-forming cell. Bone tissue is composed of osteoblast, bone-forming cells in various differentiation stages and osteoclast, bone resorption cells. In vivo, there are lots of cell-cell interactions between not only osteoblast and osteoclast but matured osteoblast and immature bone-forming cell. Those makes it difficult to identify the signaling pathway activated by E2 stimulation in vivo.
In this study, we showed E2-activated signaling pathway important to matrix mineralization of MC3T3-E1 in early differentiation stage. As next step, we are going to investigate the signaling pathway in more complex matrix mineralization processes influenced by other cells, using primary osteoblast and osteoclast.
Comments 2: The mechanisms of estrogen receptor action in the plasma membrane and cytosol are not entirely clarified, leading to potential uncertainties in the interpretation of results.
Response 2: Agree. Estrogen receptors (ERs) action you pointed is a big problem in our research and it is one of the reasons why we complete this paper once in this step. We are trying to specify the signal molecule induced by ERs through non-genomic pathway, it requires further research.
However, this study might help to clarify the research of another pathway of ERs.
Comments 3: The use of stripped serum to eliminate lipophilic hormones for studying estrogen action may oversimplify the in vivo hormonal environment, potentially affecting the relevance of the findings.
Response 3: We agree with it. Actually, we observed another hormone influenced matrix mineralization in MC3T3-E1 like E2. Moreover, E2 signal was converted to another extracellular molecule immediately in our model. It suggests that we must consider not only ERs but another receptor and signaling pathways in vivo.
Similar to response 1, we showed the existence of new signaling pathway in bone-forming cell with simplified model construction, which makes some limitation in this study. we think that this study is not enough to reveals the detail of the new signaling pathway of estradiol in bone-forming cells but is important in suggesting the unknown regulation system and function of female hormone in bone tissue.
Reviewer 2 Report
Comments and Suggestions for Authors
Review comments for ijms-2952309.
In this manuscript, the authors examined the effect of estradiol (E2) on matrix mineralization. They firstly confirmed bone anabolic effects of E2, such as mineralized nodule formation and ALP activity. Then they examined intracellular signaling of E2 and found that not only estrogen receptor-related signaling but also PKC and MAPK. In addition to signaling, they further examined supernatant-mediated bone anabolic effects of E2.
This manuscript is well-written and is of interest from the point of elucidating the mechanisms of bone anabolic effects of E2.
The followings are comments.
1. Use molar concentration for supplementary figure, as it makes easy to recognize how extent regular FBS contains E2.
2. As to the experiments of the use of collected supernatant, please describe how the authors eliminate the carry-over of E2 from first culture to second culture.
Author Response
Thank you very much for taking the time to review this manuscript. Please find the detailed responses below.
Point-by-point response to Comments and Suggestions for Authors:
In this manuscript, the authors examined the effect of estradiol (E2) on matrix mineralization. They firstly confirmed bone anabolic effects of E2, such as mineralized nodule formation and ALP activity. Then they examined intracellular signaling of E2 and found that not only estrogen receptor-related signaling but also PKC and MAPK. In addition to signaling, they further examined supernatant-mediated bone anabolic effects of E2.
This manuscript is well-written and is of interest from the point of elucidating the mechanisms of bone anabolic effects of E2.
The followings are comments.
Comments 1: Use molar concentration for supplementary figure, as it makes easy to recognize how extent regular FBS contains E2.
Response 1: Thank you for your advice. We agree with this comment. I fixed the supplementary figure 1. Please see the attachment.
Comments 2: As to the experiments of the use of collected supernatant, please describe how the authors eliminate the carry-over of E2 from first culture to second culture.
Response 2: To avoid the remaining of E2 after medium change, we washed the whole well and cell surface with collected culture itself before medium change. Rince with new culture media or PBS were considered, but we did not do that because there is a possibility that excessive wash inhibits autocrine differentiation signaling protein of MC3T3-E1 including Wnt, BMP and FGF. In Figure 7A and 9C, we showed that the signaling pathway activated by direct E2 stimulation is different from second culture from the E2-stimulated cell. Thus, it indicates our treatment was enough to eliminate the carry-over of E2.
